# Improving Skin-to-Skin Practice for babies in Kangaroo Mother Care in Malawi through the use of a customized baby wrap: A randomized control trial

**Kondwani Chavula[1]\*, Tanya Guenther[2], Bina Valsangkar[2], Victoria Lwesha[1], Gedesi Banda[1], Marte Bøe Wensaas[3], Richard Luhanga[1], Lydia Chimtembo[1], Mary V. Kinney[2], Queen Dube[4]**

**1** Save the Children, Lilongwe, Malawi, **2** Department of Global Health, Save the Children US, Washington, DC, United States of America, **3** Save the Children, Olso, Norway, **4** College of Medicine, University of Malawi, Blantyre, Malawi

\* Kondwani.Chavula@savethechildren.org

## Abstract

### Background

Complications of prematurity are a leading cause of newborn death in Malawi. Despite early adoption of Kangaroo mother care (KMC), coverage remains low and women have expressed challenges in using the traditional wrapper–*chitenje*. In 2016, a study was conducted to evaluate the acceptability and effectiveness of a customized KMC wrap in improving adherence to KMC practices among mothers.

### Methods

Mother-baby dyads (301) were randomized to receive either a customized CarePlus Wrap developed by Lærdal Global Health or a traditional *chitenje*. Enrolled mother-baby dyads were assessed in the KMC ward at 2–3 days after of admission, and then again at 7–15 days post-discharge. Topics covered included skin-to-skin practices, breastfeeding, perceptions of the wrap, and family/community support. *Chi square* tests were used to assess associations between wrap type and KMC practices. The study received ethics approval.

### Results

This study found that a customized KMC wrap is highly acceptable to women and improved skin-to-skin practices in facility-based KMC: 44% of mothers using a customized wrap reported 20 or more hours per day, compared to 33% of mothers using the traditional *chitenje*. Women using the customized wrap reported being comfortable in keeping the baby in skin-to-skin position more often than women using the chitenje (96% vs. 71%), and they were able to tie on the wrap themselves (86% vs. 10%). At the time of discharge from KMC, more women who used the customized wrap were satisfied with the wrap than those who

**Data Availability Statement:** All relevant data are within the paper and its Supporting Information files.

**Funding:** Research was funded by Lærdal Global Health and implemented by Save the Children Norway, Malawi and US. The research, including the analysis and writing of the results, was also funded by Save the Children – Saving Newborn Lives. Any opinion, finding and conclusion, or recommendation expressed in this material is that of the authors. The funders did not influence the interpretation of the results.

**Competing interests:** The authors declare that they have no competing interests.

used the traditional *chitenje* (94% vs. 56%). The customized wrap did not appear to impact other newborn practices, such as breastfeeding.

## Conclusions

This study provides evidence that a customized KMC wrap is highly acceptable to mothers, and it can contribute to better skin-to-skin practices. Use of a customized wrap may be one mechanism to support mothers in practicing KMC and skin-to-skin contact in addition to other interventions.

## Introduction

Complications from preterm birth (defined as births occurring before 37 weeks' gestation) are a leading cause of under-five mortality worldwide, accounting for an estimated 18% of global deaths [1]. Among recommendations to improve preterm birth outcomes, the World Health Organization (WHO) strongly recommends Kangaroo mother care (KMC) as part of routine care of newborn infants weighing up to 2,000g at birth, and that it should be initiated in health-care facilities as soon as infants are clinically stable [2]. The WHO defines KMC as "care of preterm infants carried skin-to-skin with the mother. Its key features include early, continuous and prolonged skin-to-skin contact between the mother and the baby, and exclusive breast-feeding (ideally) or feeding with breastmilk" [2]. KMC has been shown to be a safe and effec-tive way of reducing neonatal mortality [3]. Compared to conventional care, continuous KMC is associated with a reduced mortality risk of 40% at the time of discharge or at 40–41 weeks postmenstrual age and improves other outcomes, including severe infection/sepsis and weight gain [3]. Specific to preterm birth, KMC has been shown to lead to a large, cause-specific decrease of 51% (95% CI 18–71% reduction) in neonatal deaths with birth weight of <2,000g [4].

The success of KMC depends, in part, on the ability of mothers and caregivers to practice continuous skin-to-skin contact, whereby infants are placed vertically on the chest between the mother's breasts below her clothes. Practicing KMC, as recommended, can be difficult for women [5]. Pain and fatigue are common barriers in low- and middle-income countries, with women indicating that the baby is too difficult to hold; that they experience physical discom-fort on the chest or back, and that it is difficult to maintain the proper position of the infant, especially when the mother is sleeping. Despite the challenges they face to adhere to the recom-mended practices, KMC is acceptable to mothers [5, 6]. Top enablers include experiential fac-tors, such as mother-infant attachment and ease of practice, as well as support from family, friends, and other mothers, especially to give the mother a break from practice or to allow her to deal with other tasks, such as childcare and housekeeping [5]. Some studies have shown that the level of adherence of post-discharge KMC practice remain high [7, 8](Ghana, India), yet studies from Malawi have found lower levels of adherence post-discharge [9]. Studies assessing women's perceptions in Malawi on KMC following discharge have found women being over-whelmed with responsibilities at home, developing anxiety and fatigue discouraging them from continuing KMC, and experiencing financial difficulties, lack of support, and stigma, ultimately resulting in following unhealthy, traditional care practices of LBW babies [10–13]. Mothers in Malawi have also identified positioning of the infant and discomfort among the main barriers to KMC practice when using the traditional wrap called a *chitenje*, which is a piece of fabric normally two meters long by one meter wide [10].

Malawi was an early adopter of KMC, initiating the intervention in 1999, and then it scaling up KMC services in all central and district hospitals in 2005 after national KMC guidelines were developed [14]. By 2014, 77% of hospitals reported that they provided inpatient KMC services, with some challenges noted around functionality and documentation [14]. In 2017, an estimated 15,995 babies were initiated on KMC nationally, representing about 22% of expected cases nationwide (calculated as 10% of expected live births) and 44% of preterm or LBW babies reported in facilities [15]. The Government of Malawi has reaffirmed its commitment to improve newborn health and to reduce mortality and morbidity by implementing KMC [16].

To improve uptake of KMC, in 2015, Lærdal Global Health launched an ergonomic KMC baby carrier (CarePlus Wrap) designed to make it easier for mothers to practice KMC. Designed with the intention of local production at a low cost, each wrap is accompanied by pictorial images on how to use it. Lærdal Global Health also hoped that mothers would take the wraps home and then return them at scheduled KMC follow-up visits, so that the wraps could be recycled and used by other mothers. The wrap was pre-tested in Malawi and Tanzania, with health workers trained on how to use the wrap and how to counsel mothers on how to maintain the baby in skin-to-skin position. The pre-test demonstrated that the customized wrap was perceived to be safer, more comfortable, easier to use, and more acceptable to wear (for mothers and fathers) than the traditional wrap [17].

Currently, there are limited data from low-resource settings on the duration of skin-to-skin practice [18] and on KMC practice with different types of wraps [19]. We conducted this study to address the evidence gap on the acceptability and effectiveness of a custom KMC wrap on adherence to skin-to-skin practices in Malawi. It responds to the call for implementers to study the effectiveness of user-centric designs for promoting KMC [5]. Specific objectives include: (1) to assess acceptability of the wrap and adherence to skin-to-skin practice with the introduction of a KMC wrap in selected health facilities, and (2) to assess skin-to-skin practice in the communities, post-discharge from facility-based KMC.

## Methods

### Study design

This was implementation research, with a randomized control trial (RCT) design. Mother-baby dyads meeting eligibility criteria and providing informed consent were randomly assigned to receive either the customized CarePlus KMC Wrap (intervention group) or a traditional *chitenje* (control group). All facility staff involved in providing KMC received a one-day refresher training on KMC, including continuous counselling for the mothers. As per standard protocol, all mothers whose babies were targeted for facility-based KMC were oriented on how to practice KMC, including how to practice skin-to-skin contact, how to care for their small babies (covering breastfeeding, expressing breast milk, cup feeding, hygiene, and the importance of skin-to-skin position) when the baby was initiated on KMC. All mothers received specific instruction on how to use the type of wrap they received: mothers receiving the CarePlus Wrap were shown how to tie and use the CarePlus Wrap (Fig 1), while mothers receiving the *chitenje* were shown how to tie and use the *chitenje*.

### Study sites

The study was conducted in three large hospitals in the southern region of Malawi: Machinga District Hospital, Thyolo District Hospital, and Queen Elizabeth Central Hospital (Blantyre district) (Table 1). All three facilities had previously established KMC services.

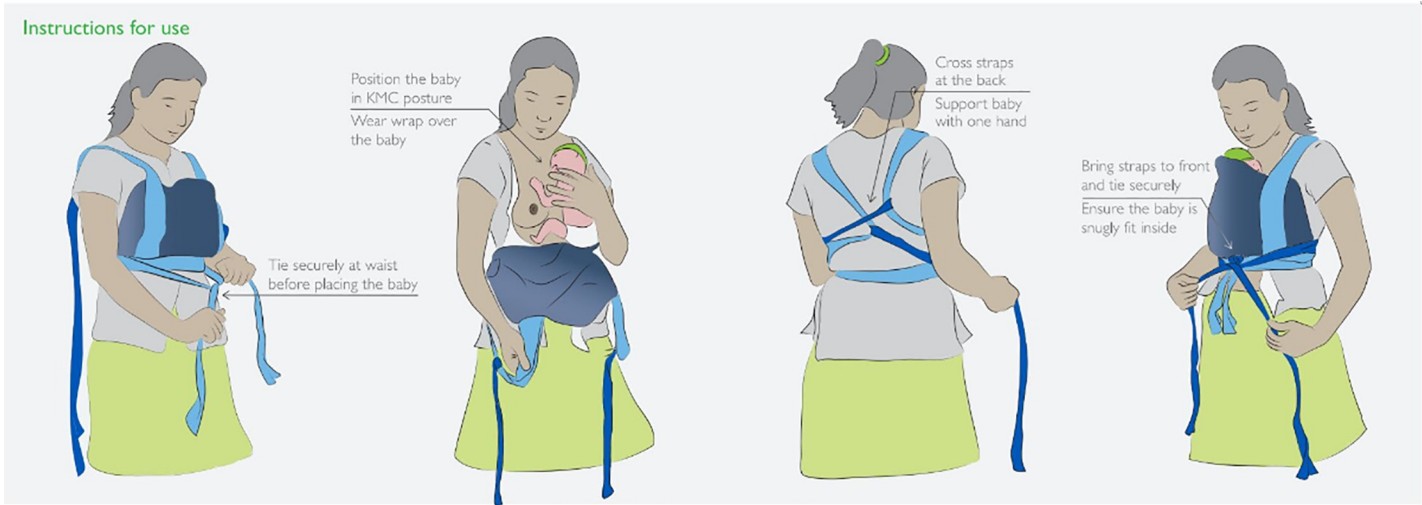

**Fig 1. Instructions for CarePlus Wrap use.**

## Study population and sample size

We targeted a sample size of 280 mother-baby dyads (140 per arm) for the primary outcome of interest–weight gain–to detect a minimum 20% difference between groups (S1 Appendix). We evaluated all admitted mother-baby dyads, included multiple births, at the three hospitals for study eligibility within 24 hours of admission to KMC. For enrollment, we excluded babies who had known neurological problems, had major complications, had birth weights of less than 800g, were aged ≥7 days at admission to KMC, had their mother die during labor, were referred out for higher level care, or lived more than 10km from the study hospital. Mothers and their babies eligible for facility-based KMC (birth weight less than 2,000g) at these three hospitals, who provided informed consent to take part in the study, were randomly assigned to either the intervention arm or the control arm.

## Data collection

Mother-baby dyads meeting eligibility criteria and providing informed consent were randomized to receive either the CarePlus wrap or a traditional chitenje printed with KMC messages. Eligible participants admitted to KMC who had consented to take part in the study were allocated to intervention or control groups in sequential order. The initial allocation of the respondent to treatment or control group was determined at random, and the team doing the

**Table 1. Information about study sites in 2016.**

| Facility Name | Level | Deliveries (#) | KMC beds (#) | KMC patients per month (#) | Support staff in KMC unit |
|---|---|---|---|---|---|
| Queen Elizabeth Central Hospital | Tertiary-level government district hospital | 12150 | 35 | ~107 cases | Trained patient attendants Midwives Pediatrician |
| Machinga district hospital | Government district hospital | 5576 | 4 | 32 cases | Midwives |
| Thyolo district hospital | Government district hospital | 5103 | 8 | 17 cases | Clinical officer Midwives Patient attendant |

Source: Number of deliveries for Queen Elizabeth Central Hospital from the Obstetrics Department and number of deliveries for Machinga and Thyolo district hospitals extracted from DHIS2; number of beds from observation; number of patients per month from registers; support staff from observations.

allocation were not aware of which study arm the respondent will be assigned, followed by assigning second recruitment to the alternate group and so on. The alternation was repeated each day there is a new client until the required sample size was obtained.

Data were collected from May to December 2016. Data collection teams comprised of research officers, field workers, and clinical observers were trained on the study protocol for four days by the study's principal investigator and Save the Children staff. Using a pretested, structured questionnaire (S2 Appendix), data collectors surveyed mothers about their skin-to-skin practice, breastfeeding, perceptions of using the wrap (CarePlus or *chitenje*), and family/community support at 2–3 days after admission to KMC, at discharge, and 7–15 days post-discharge. Data collectors extracted information from newborn case files, maternity registers, and KMC registers. All data were captured using tablets programmed with CS-PRO.

## Outcomes and data analysis

The primary outcome variable was the average rate of weight gain among babies enrolled in KMC; results are not included in this paper (explanation in S1 Appendix). For this paper, we report on the secondary study outcomes including maternal reports of wrap acceptability, reported duration of skin-to-skin contact, feeding support, and family and other social support (Table 2). We consider overall results by control and by intervention group. Chi square tests were used to assess differences between study arms and p-values less than 0.05 were considered statistically significant. We conducted analyses at two time points, in-facility and post-discharge. All analyses were carried out in Stata® 12 [StataCorp LP, Texas, United States].

**Table 2. Description of study outcomes explored in this study.**

| Outcome area | Metrics |
|---|---|
| Acceptability of the wrap | Proportion of women receiving a CarePlus wrap would recommend it to other mothers (measured on third day in facility and at time of discharge)<br>Proportion of mothers expressing preference for the CarePlus wrap over the new chitenje measured at time of discharge and post discharge |
| | Proportion of women who reported the wrap they received was: a) comfortable to keep baby in skin-to-skin; b) easy to tie by herself; b) comfortable to use while breastfeeding; c) comfortable to use while sleeping; d) keeps baby in secure position; and e) acceptable for male family members to use (measured at time of discharge) |
| Adherence to skin-to-skin practices | Proportion of babies whose mothers reported (1) practicing any skin-to-skin post discharge and (2) practicing skin-to-skin more than half the day and more than half the night post-discharge (measured at 7–10 day visit)<br>Proportion of mothers reporting (1) daily duration, (2) day time, (3) night time of skin-to-skin practice while in facility and post discharge |
| Feeding practices | Proportion of babies whose mothers reported (1) breastfeeding and (2) required use of cup and spoon to support feeding at the time of discharge |
| Family and Social Support | Proportion of women reporting receiving support from family in (1) facility and (2) post discharge<br>Proportion of women reporting receiving support from community members including friends and women's groups post discharge. |
| Scalability of the CarePlus wrapper | Proportion of women receiving a CarePlus wrap indicating willingness to return the wrapper<br>Proportion of women receiving a CarePlus wrap expressing willingness to use a recycled wrap.<br>Proportion of women receiving a CarePlus wrap reporting that they would pay a deposit for the wrap or buy one.<br>Reported ability of hospital tailors to produce the wrappers locally with availability of the right materials. |

### Ethics

Ethical approval for the study was received in May 2016 from the College of Medicine Research and Ethics Committee (COMREC) national bioethics committee (Reference number: P.11/15/1835). Written informed consent was obtained from every mother/baby dyad who agreed to participate in the study. Mothers under 18 years were excluded from the study.

## Results

### Overview of sampling and study participants

A total of 581 babies were assessed for eligibility (Fig 2), of which 318 met inclusion criteria; mothers of 301 provided informed consent, and they were enrolled in the study (152 in the CarePlus Wrap group, 149 in the *chitenje* group). The large (n = 75) number of ineligible babies in the "other" category were babies with birthweight >2000g. Of those enrolled, 85% (129) of the intervention group and 77% (114) of the control group were followed through 7–15 days post-discharge. There were no significant differences between study arms at enrollment in maternal background or health characteristics, or in characteristics of babies (S3 Appendix). More than one third of babies (36%) were less than 1,500g at initiation; about 20% were between 1,800g and 2,000g. Most babies were preterm, with close to two thirds estimated to be between 32 and 36 weeks gestation. Less than 15% of babies were documented to have experienced other complications at birth (in addition to preterm/low birth weight); asphyxia was the most commonly reported complication. The average age at admission to KMC was 1.5

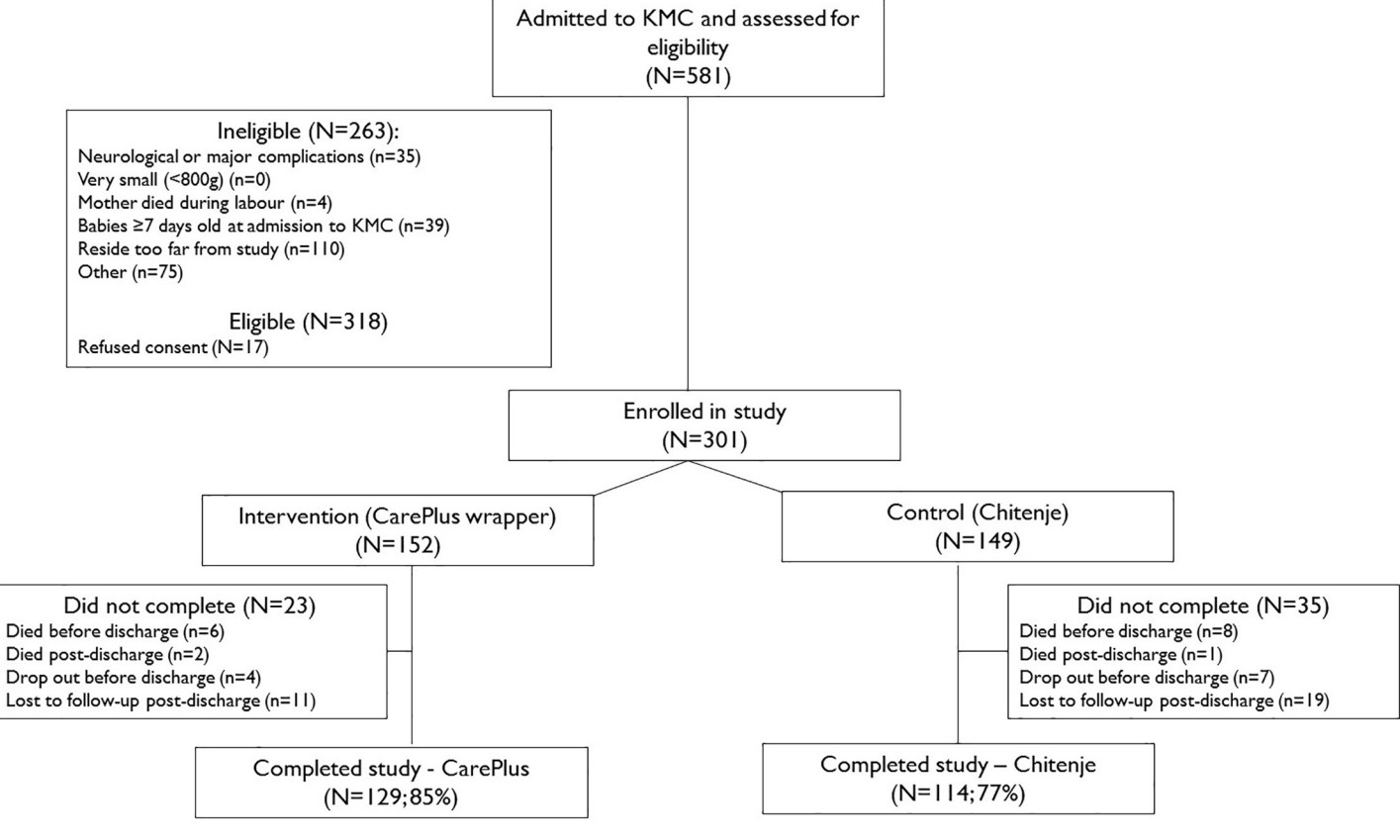

**Fig 2. Selection flow diagram.**

days, and it was similar across study arms. Average duration of stay in facility KMC was 9.3 days, with babies in the intervention arm staying slightly longer than control babies; the difference was not statistically significant. About 20% of babies experienced major complications and needed special care (e.g., severe respiratory problems, severe infections) while in facility KMC, with intravenous therapy and phototherapy being the most common treatments received and no difference between study arms (4% in both arms).

## Outcomes

**Acceptability.** Women expressed higher levels of acceptability for a customized wrap than the traditional wrap (Fig 3). Within the first three days of admission, a significantly higher proportion of women using the CarePlus Wrap reported being comfortable keeping the baby in skin-to-skin position (96% vs. 71%), being able to tie on the CarePlus Wrap themselves (86% vs. 10%), being comfortable to breastfeed while using the CarePlus Wrap (43% vs. 12%), and that the CarePlus Wrap would be acceptable for use by fathers and male family members (93% vs. 43%) (Fig 2). Women using the Care Plus Wrap reported less time to become comfortable using the wrap, with 72% comfortable within the first day compared to 40% of those using the *chitenje* (*p* = 0.00). In terms of satisfaction with their wrap, 94% of women who used the CarePlus Wrap were very satisfied with it, compared to 56% of those who used the traditional *chitenje* (*p* = 0.00).

Satisfaction with the CarePlus Wrap remained high 7–10 days post-discharge. Nearly all women in the intervention arm (95%) reported that the wrap was suitable for husbands or other male family members to use, and that they would recommend the wrap to other mothers (91%). When asked about preference between the CarePlus Wrap and the *chitenje* for practicing KMC, women using the CarePlus stated they preferred the CarePlus Wrap. Regardless of the wrap used, women reported satisfaction with their baby's progress on KMC (94%), and they would recommend KMC to others (99%).

**Adherence to skin-to-skin practice.** Reported daily duration of skin-to-skin practice while in facility was significantly higher among women using the CarePlus Wrap (Table 2). Among women using the CarePlus Wrap, 44% of women reported more than 20 hours skin-to-skin contact per day; 6% reported 10 hours or less of skin-to-skin contact. In comparison, 33% of women using the traditional *chitenje* reported 20 hours or more of skin-to-skin contact per day, and 19% reported 10 hours or less per day. Duration of skin-to-skin contact was high for both study groups, with 39% reporting 20 or more hours of daily practice and 45% reporting between 11 and 19 hours. Women using the CarePlus Wrap were also more likely to report

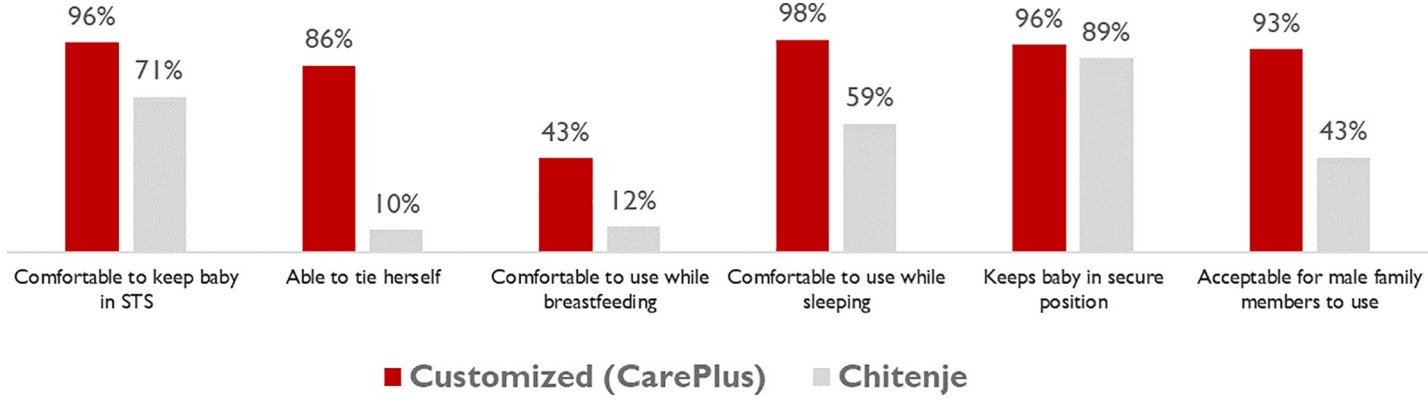

**Fig 3. Differences in the acceptability of wraps while in facility-based KMC by type of wrap received (day 3).**

using the wrap for nearly all the day (54% vs. 41%), and nearly all or more than half of the night than women using the traditional wrap (97% vs. 83%). Among the 120 women who reported practicing nearly all day in facility, most reported practicing all night ($n$ = 111, 93%), with no significant difference between groups (results not shown).

Reported daily duration of post-discharge skin-to-skin practice remained high (87% of all those who completed the study), without a significant difference between study arms. Among these women, nearly all reported daily skin-to-skin practice (92%). Duration of practice varied, with nearly half of women reporting skin-to-skin practice as "more than half the day" (47%), and the remaining women indicating practice either as "nearly all the day" (28%) or "some of the day" (23%), with no significant difference between groups (Table 3). Among the 67 women who reported practicing nearly all day at the post-discharge visit, most reported practicing all night ($n$ = 62, 93%), with no significant difference between groups.

**Feeding practices.**   Nearly all babies were able to breastfeed while in facility, regardless of wrap type (Table 4, 98.4% for the CarePlus Wrap vs. 100% for the *chitenje*). Nearly all mothers mentioned receiving support from health facility staff for expressing breastmilk (92%), for feeding position (91%), and for reminders for feeding (87%); no significant differences were noted between study arms. Nearly all babies were also able to breastfeed post-discharge, regardless of wrap type (97.7% vs. 97.4%).

**Family and social support.**   Reported levels of family and social support while in facilities were not significantly different between women using the CarePlus Wrap and women using the traditional *chitenje* (Table 5). Overall, 63% of women reported that another family member was trained in KMC while in facility, usually the mother, sister, or mother-in-law. Only 1% of women reported that their husbands received training in KMC at the facility. Nearly all women (90.9%) reported receiving some form of help from family members while they were in facility KMC, with the most common support being provision of food, followed by help holding the baby in skin-to-skin position and caring for children at home. Maternal grandmothers, husbands, and sisters were the most commonly mentioned family members providing this support.

Following discharge from facility, most women (>85%) reported receiving support from their families, with helping to hold the baby in skin-to-skin position being the most frequently reported form of support, followed by providing food and helping with household chores. While no significant differences were noted between study arms in terms of level of family support, the type of support did vary, with significantly more women using the CarePlus Wrap reporting family support for household chores. Mothers, husbands, and sisters were the most commonly mentioned family members providing this support, with no significant differences between women using the CarePlus wrapper and women using the *chitenje*. Levels of support from outside the family were also similar between study groups; provision of food and help with household chores were the most commonly reported forms of external support. More than one third of women reported receiving support from community members, with friends and women's groups being the most commonly mentioned sources of support.

**Scalability of the CarePlus Wrap.**   We conducted a preliminary, informal exploration of the feasibility of local production and the willingness of women to purchase or return the CarePlus wrap. We found that tailors can produce the CarePlus wrap locally, and that more than half of women reported being willing either to pay a deposit or to buy a wrap (Box 1). The major challenge, however, is how to scale up and sustain production of quality wraps and procurement of the customized wrap within the health care system. A customized wrap may be acceptable and increase skin-to-skin adherence, but investigation on how to include and sustain this new commodity into the supply chain is necessary.

**Table 3. Reported skin-to-skin practices while in facility-based KMC and post-discharge.**

| Skin-to-skin practices in <u>facility KMC</u> | Study Group CarePlus Wrap (*n* = 126) | *Chitenje* (*n* = 115) | *p*-value | Total(*N* = 241) |
|---|---|---|---|---|
| **Usual amount of time in skin-to-skin contact during daytime** | | | **0.00** | |
| Nearly all the day | 54.0 | 40.9 | | 47.7 |
| More than half the day | 43.9 | 45.2 | | 44.0 |
| Some of the day (less than half) | 3.2 | 13.9 | | 8.3 |
| **Usual amount of time in skin-to-skin contact during nighttime** | | | **0.00** | |
| Nearly all the night | 52.4 | 49.6 | | 51.0 |
| More than half the night | 44.4 | 33.0 | | 39.0 |
| Some of the night (less than half) | 3.2 | 17.4 | | 10.0 |
| **Reported hours/day baby usually placed in skin-to-skin contact** | | | **0.03** | |
| 20 or more hours | 44.4 | 33.0 | | 39.0 |
| 11–19 hours | 46.0 | 44.4 | | 45.2 |
| 5–10 hours | 5.6 | 18.3 | | 11.6 |
| 4 hours or less | 0.8 | 0.9 | | 0.8 |
| *Don't know* | 3.2 | 3.5 | | 3.3 |
| **Skin-to-skin practices <u>post-discharge</u>** | CarePlus Wrap (*n* = 126) | *Chitenje* (*n* = 115) | *p*-value | **Total (*N* = 243)** |
| **Baby in skin-to-skin position (observed)** | 52.7 | 48.3 | 0.49 | 50.6 |
| **Still practicing skin-to-skin contact (reported)** | 86.1 | 88.6 | 0.55 | 87.2 |
| **Practicing skin-to-skin contact every day (for those practicing)** | 94.6 | 89.5 | 0.22 | 92.2 |
| **Usual amount of time in skin-to-skin contact during daytime**[*] | | | 0.66 | |
| Nearly all the day | 28.8 | 27.7 | | 28.3 |
| More than half the day | 46.0 | 48.5 | | 47.2 |
| Some of the day (less than half) | 24.3 | 20.8 | | 22.6 |
| Not at all | 0.9 | 3.0 | | 1.9 |
| **Usual amount of time in skin-to-skin contact during nighttime**[*] | | | 0.07 | |
| Nearly all the night | 59.5 | 69.3 | | 64.2 |
| More than half the night | 29.7 | 18.8 | | 24.4 |
| Some of the night (less than half) | 6.3 | 10.9 | | 8.5 |
| Not at all | 4.5 | 1.0 | | 2.8 |

[*]Data available for 212 cases (111 intervention and 101 control) who were still practicing skin-to-skin contact at the post-discharge interview. Note: the categories for "usual amount of time" (nearly all the day/night, more than half the day/night, some of the day/night) were not specified, and they were left to the respondents to determine.

**Table 4. Reported feeding practices while in facility-based KMC and post-discharge.**

| Maternal report of feeding practices in <u>facility KMC</u> | Study Group CarePlus Wrap (*n* = 126) | *Chitenje* (*n* = 115) | *p*-value | **Total(*N* = 241)** |
|---|---|---|---|---|
| **Baby able to breastfeed** | 98.4 | 100.0 | 0.18 | 99.2 |
| **Type of feeding** | | | 0.38 | |
| Exclusive breastmilk | 92.1 | 95.7 | | 93.8 |
| Predominantly breastmilk | 1.6 | 1.7 | | 1.7 |
| Partial breastmilk | 6.4 | 2.6 | | 4.6 |
| **Maternal report of feeding practices <u>post-discharge</u>** | CarePlus Wrap (*n* = 126) | *Chitenje* (*n* = 115) | *p*-value | **Total (*N* = 243)** |
| **Baby able to breastfeed post-discharge** | 97.7 | 97.4 | 0.88 | 97.5 |
| **Baby fed anything other than breastmilk since discharge** | 3.1 | 8.8 | 0.06 | 5.8 |

**Table 5. Levels of reported family and social support for KMC.**

| Maternal report of support <u>in facility KMC</u> | Study Group | | | Total (N = 241) |
|---|---|---|---|---|
| | CarePlus Wrap (n = 126) | Chitenje (n = 115) | p-value | |
| **Other family member trained on KMC at facility** | 60.3 | 65.2 | 0.43 | 62.7 |
| **Received help from family member (either at home or in facility)** | 91.3 | 90.4 | 0.82 | 90.9 |
| **Type of help provided** | | | | |
| Provided food | 85.7 | 89.6 | 0.37 | 87.6 |
| Held baby in skin-to-skin position | 38.9 | 39.1 | 0.97 | 39.0 |
| Cared for children at home | 32.5 | 43.5 | 0.08 | 37.8 |
| Provided moral support | 32.5 | 34.8 | 0.71 | 33.6 |
| Other | 15.1 | 11.3 | 0.39 | 13.3 |
| **Maternal report of support post-discharge** | CarePlus Wrap (n = 126) | Chitenje (n = 115) | p-value | Total (N = 243) |
| **Any family member provided help since returned home** | 88.4 | 86.8 | 0.72 | 87.7 |
| **Type of help provided** | | | | |
| Held baby in skin-to-skin position | 81.4 | 82.5 | 0.83 | 81.9 |
| Provided food | 78.3 | 71.9 | 0.25 | 75.3 |
| Household chores | 79.1 | 66.7 | **0.03** | 73.3 |
| Cared for children at home | 52.7 | 43.9 | 0.17 | 48.7 |
| Provided moral support | 60.5 | 53.5 | 0.27 | 57.2 |
| Other | 0.8 | 4.4 | 0.07 | 2.5 |
| **Anyone outside of family providing support** | 39.5 | 35.1 | 0.48 | 37.5 |
| Type of support provided | | | | |
| Provided food | 30.2 | 20.2 | 0.07 | 25.5 |
| Household chores | 27.9 | 18.4 | 0.08 | 23.5 |
| Cared for children at home | 6.2 | 6.1 | 0.98 | 6.2 |
| Provided moral support | 20.9 | 18.4 | 0.62 | 19.8 |
| Held baby in skin-to-skin position | 4.7 | 5.3 | 0.83 | 4.9 |
| Other | 3.1 | 5.3 | 0.40 | 4.1 |

## Discussion

We set out to determine the adherence to KMC practices as well as the acceptability and effectiveness of a customized KMC wrap versus a traditional wrap. The findings indicated that use of the CarePlus Wrap facilitated extended duration of skin-to-skin practice while in facility, with significantly more women in the CarePlus group reporting continuous skin-to-skin contact than women in the *chitenje* group. While there was no significant difference in KMC between groups following discharge, the duration of skin-to-skin practice post-discharge remained high. Reported breastfeeding practices and family and other social support did not differ between the groups post-discharge.

We found that women reported higher levels of satisfaction with the customized wrap than the traditional wrap; however, we did not test preference between wraps; rather, the acceptability of a customized wrap. Feasibility research in Nepal assessing the CarePlus Wrap compared to a traditional wrap found women preferring the customized wrap [20, 21]. A study from Indonesia exploring perceptions of using different types of KMC wraps found no difference in preference between wraps [19]. Besides these studies, other related KMC studies do not compare wraps, but rather they use customized wraps in all cases or do they not mention the type of wrap used [3, 22–24]. Some countries and facilities have institutionalized the use of a customized wrap, without testing its impact; e.g., South Africa uses the Kalafong KMC Thari

## Box 1. Scalability and local production of the customized wrap

Alongside the study, we explored local production and the feasibility of implementing a recycling scheme for the wraps and willingness to pay for the wraps. Two areas of feasibility were explored: (1) producing the customized wrap by training and equipping existing hospital-based tailors; and (2) the willingness of women to return used wraps, use pre-used wraps, and pay for wraps.

### How we assessed scalability?

Technical information (in simple terms) was shared with five hospital-based tailors from the three study sites to enhance their understanding about preterm birth and LBW babies. After the training, the matron from each facility assessed each wrap produced by the local tailor to determine if it was produced to the required specification. Women were asked, during the KMC wrap study data collection, about their willingness to return the CarePlus Wrap to the facility, to buy a wrap, and how much they would be willing to spend.

### What we found?

Training of tailors showed that it is possible to produce the wraps locally with availability of right materials. The cost of the local materials required to produce the wrap was estimated to be approximately US$2, this included the cost of the fabric only (tailors were already commissioned by the hospitals). Based on the assessment of time to produce each wrap and time already allocated for current tailoring responsibilities, each tailor would be capable of producing approximately two wraps per day.

Close to two thirds of women in the CarePlus group indicated that they would be willing to return the wrap, and 70% expressed willingness to use a recycled wrap. Similarly, about 64% of women would either pay a deposit for the wrap or buy one (63%) for about US$2 (the estimated cost of local production).

### What we recommend?

Wraps can be produced locally, but it will require materials and allocated time for the hospital tailors to produce the wraps. To meet initial demand for wraps at each hospital in a timely manner, it may be necessary to contract additional local tailors to reach an adequate starting stock of wraps. Assuming that about 60% of wraps would be returned and that wraps would be lost due to wear and tear over time, hospital tailors would need to produce replacement wraps continually. The Ministry of Health and implementing partners will need to develop and test approaches for locally producing and financing a customized wrapper. The Ministry of Health should consider testing how to incorporate a customized wrap into district health system planning processes, e.g., include a customized wrap in the procurement catalogue and district expenditure plans for sustained availability at scale.

Wrap [25] and Bangladesh uses the KMC Binder [26]. Although we did not conduct a comparative analysis in terms of texture or type of wraps with these studies, countries and facilities should consider including details on the type of wrap used in future research. Different models of wraps should be tested in different contexts to determine preference and impact on adherence to KMC practices.

Our study found high daily duration of skin-to-skin practice overall, with over 84% of women reporting at least 10 hours or more in facility. A mapping of related RCTs in the 2016 Cochrane Review of KMC identified 16 studies that reported on the mean daily duration of skin-to-skin contact and found that the mean daily hours of KMC ranged from 10 minutes to 17 hours per day [3, 18]. Among these studies, Watkins et al. found none from sub-Saharan Africa, and the method of recording hours of skin-to-skin was rarely reported as observation, self-report, or other methods [18]. In addition to the studies included in the Cochrane review, an observational study in Uganda found a mean of only three hours of skin-to-skin contact over the first week of life [18]. The feasibility study from Nepal found that all reported practicing skin-to-skin contact was for 10 hours or more daily in the first month, regardless of wrap (longer duration was reported among mothers using the customized wrap, though this was not statistically significant) [21]. As a strong recommendation, the most recent WHO guidelines indicate that newborns weighing 2,000g or less at birth should be provided "as close to continuous Kangaroo mother care as possible" and "intermittent" KMC if continuous care is not possible. However, the guidelines do not define the minimum duration per day that KMC should be practiced [2]. As studies focus on the number of hours, there need to be greater efforts to achieve continuous KMC, which implies no interruption of skin-to-skin contact during KMC practice, as per the recommendation. As shown in our study, when asked about duration differently, only half of women reported practicing KMC "nearly all day" or "nearly all night," and those women who were practicing all day were mostly practicing all night too. The framing of the question to determine continuous needs to be examined.

Previous studies have identified other factors beyond the wrap influencing KMC practices, including inadequate staffing levels and capacity, limited provision of information about KMC to mothers during antenatal care/shortly after birth, inadequate equipment and supplies to support mothers' adherence to recommended KMC practices, dedicated space, addressing the pain and fatigue of mothers in facilities and support to the mothers post-discharge, and weak recording and reporting on KMC [5, 10–12, 15]. These other challenges also warrant further examination to improve the quality of KMC implementation in Malawi.

Our study found that mothers practicing KMC reported receiving social support from family members and the community. The study also found little engagement of husbands in the support of skin-to-skin, contact, regardless of the wrap used. The level of family and social support has been previously identified as an important enabler for KMC practices in Malawi [6, 11, 27]. Though there is not enough evidence on the involvement of fathers in increasing adherence to KMC [3], several studies report that stigma around male involvement in child care prevents KMC uptake [27, 28]. A more family-centered care approach to KMC may increase initiation and continued practice, but it would need to engage men in a way that also empowers women [28]. It should be noted that at the time of the study, an intensive SBCC campaign around preterm birth and KMC was being implemented in two of the districts, and it could have contributed to this result. The campaign specifically aimed to reduce stigma, making women more comfortable practicing KMC openly in the community, regardless of which type of wrap was being used [29].

This study adds to the literature on KMC practices following discharge [3]. At the post-discharge visit in the community, 87% of women reported still practicing either intermittent or continuous skin-to-skin KMC, with three quarters indicating they were practicing more than

half of the day. This finding is consistent with other studies. The Nepal feasibility study found that all women reported continuing KMC at home at the 28-day postnatal visit [21]. A descriptive study from India reported that 82.5% of mothers were continuing with skin-to-skin contact at the 45-day post discharge visit [7]. A longitudinal study from Ghana found that 99.5% of mothers were still practicing either intermittent or continuous KMC at the first follow-up visit (one week post-discharge); the proportion did not change significantly in the first month [8]. A recent study in Malawi also found that nearly all mothers (99.2%) reported practicing skin-to-skin contact in the community following discharge [30]. Reported skin-to-skin practice at 30 days post-discharge was 67.8% for more than half of the day and 56.8% for more than half the night.

## Limitations

This study had some important limitations. Information on skin-to-skin practices was based on maternal report; it is possible that women may have misreported the duration of skin-to-skin due to social desirability bias. We do not have adequate data to compare daily direct observations with self-reported values for duration of skin to skin due to limited staffing to carry out direct observations. The novelty of a non-traditional, unfamiliar wrap may have contributed to increased adherence and satisfaction with the CarePlus Wrap. Our study, however, suggests that practical parameters, such as ease of use and comfort, were likely drivers of mothers' acceptance of the CarePlus Wrap in the intervention arm.

Our inclusion criteria prevented enrollment of the babies who were most vulnerable (i.e., had lost their mothers, were admitted at ≥7 days to KMC, or were referred out for higher level care). Fifty-eight cases did not complete the study (23 in the intervention arm and 35 in the control arm). The slightly higher level of drop-out among women using the traditional wrapper may have introduced a bias in our results related to skin-to-skin adherence and other practices that we are not able to determine. Data were missing on vital status for 30 enrolled cases, who were lost to follow-up after discharge from facility (10%). As such, it is possible that more babies originally enrolled in the study may have died while in the community after discharge from facility. Further, the study was conducted in two districts and results on acceptability may not be generalizable to Malawi as a whole.

Our implementation experience indicated that follow-up visits by study teams 7–10 days after discharge from facility KMC proved difficult to achieve in the Malawi context with the limited research resources. Only 64% of babies discharged alive received a follow-up visit between 7 and 10 days (67% of the intervention arm and 62% of the control arm). Therefore, for this study, we extended the follow-up period to up to 15 days after discharge. Extending the timeframe aligned with the Malawi KMC guidance for follow-up every two weeks until the baby is 2,500g [31]. Future studies may need to allow a larger time window or to employ additional staff.

The findings have not been published for weight gain due to limitations in data quality and capture (S1 Appendix). Prior systematic reviews looking at the effect of KMC on a multitude of health outcomes have found weak linkages between weight gain and KMC [3].

Finally, the study was not designed to compare across sites. Future research should consider study designs allowing for facility comparisons to understand variation in uptake and quality of KMC practices by facility and enabling consideration of factors beyond the wrap better.

## Conclusion

This study looked the acceptability and effectiveness of introducing a customized KMC wrap to improve skin-to-skin practices. We found that a customized KMC wrap is highly acceptable

to mothers, and it contributes to improved skin-to-skin practices while in facility KMC. Women using the customized wrap were more satisfied with KMC, and they practiced skin-to-skin contact for more hours every day while in the facility. A customized wrap may be one mechanism to support mothers in practicing skin-to-skin contact as part of KMC.

## Supporting information

**S1 Appendix. Sample size calculation.**
(DOCX)

**S2 Appendix. Data collection tools.**
(DOCX)

**S3 Appendix. Background characteristics of enrolled mothers.**
(DOCX)

**S1 Table. Data set.**
(DTA)

## Acknowledgments

We would like to thank the Ministry of Health for supporting the implementation of this study, especially the District Health Management Teams from Thyolo and Machinga, the Director of Queen Elizabeth Central Hospital, and the staff engaged in the study. This study would not have taken place without the mothers and families agreeing to participate. Steve Wall, Lara Vaz and Lauren DuComb from Save the Children US provided helpful inputs and review on the manuscript. We also appreciate the review of reports and data by colleagues from Lærdal Global Health, who funded this study.

## Author Contributions

**Conceptualization:** Tanya Guenther, Bina Valsangkar, Marte Bøe Wensaas, Lydia Chimtembo, Queen Dube.

**Data curation:** Kondwani Chavula, Tanya Guenther.

**Formal analysis:** Kondwani Chavula, Tanya Guenther.

**Funding acquisition:** Bina Valsangkar, Marte Bøe Wensaas.

**Investigation:** Kondwani Chavula, Richard Luhanga, Queen Dube.

**Methodology:** Kondwani Chavula, Tanya Guenther, Bina Valsangkar, Richard Luhanga, Queen Dube.

**Project administration:** Victoria Lwesha, Richard Luhanga, Lydia Chimtembo, Queen Dube.

**Resources:** Tanya Guenther, Bina Valsangkar, Marte Bøe Wensaas.

**Software:** Kondwani Chavula, Tanya Guenther.

**Supervision:** Bina Valsangkar, Victoria Lwesha, Gedesi Banda, Queen Dube.

**Validation:** Kondwani Chavula, Tanya Guenther, Bina Valsangkar, Mary V. Kinney.

**Visualization:** Kondwani Chavula, Tanya Guenther, Victoria Lwesha, Gedesi Banda, Mary V. Kinney.

**Writing – original draft:** Kondwani Chavula, Tanya Guenther, Mary V. Kinney, Queen Dube.

**Writing – review & editing:** Kondwani Chavula, Tanya Guenther, Bina Valsangkar, Victoria Lwesha, Gedesi Banda, Marte Bøe Wensaas, Richard Luhanga, Lydia Chimtembo, Mary V. Kinney, Queen Dube.

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
