## [Decision Letter · Decision Letter 0]

19 Dec 2019

PONE-D-19-24170

Improving skin-to-skin practice for babies in Kangaroo Mother Care in Malawi through the use of a customized baby wrap

PLOS ONE

Dear Mr Chavula,

Thank you for submitting your manuscript to PLOS ONE. After careful consideration, we feel that it has merit but does not fully meet PLOS ONE’s publication criteria as it currently stands. Therefore, we invite you to submit a revised version of the manuscript that addresses the points raised during the review process.

The reviewers' comments are complementary and all should be addressed.

We would appreciate receiving your revised manuscript by Feb 02 2020 11:59PM. To enhance the reproducibility of your results, we recommend that if applicable you deposit your laboratory protocols in protocols.io, where a protocol can be assigned its own identifier (DOI) such that it can be cited independently in the future. For instructions see: http://journals.plos.org/plosone/s/submission-guidelines#loc-laboratory-protocols

We look forward to receiving your revised manuscript.

Kind regards,

Jennifer Yourkavitch

Academic Editor

PLOS ONE

Journal Requirements:

1.

2.

We note that you have indicated that data from this study are available upon request. PLOS only allows data to be available upon request if there are legal or ethical restrictions on sharing data publicly. For information on unacceptable data access restrictions, please see http://journals.plos.org/plosone/s/data-availability#loc-unacceptable-data-access-restrictions.

3. Please provide additional details regarding participant consent. In the ethics statement in the Methods and online submission information, please ensure that you have specified what type of informed consent you obtained (for instance, written or verbal, and if verbal, how it was documented and witnessed). If your study included minors, state whether you obtained consent from parents or guardians. If the need for consent was waived by the ethics committee, please include this information.

Additional Editor Comments (if provided):

Please address the reviewers' comments regarding the methodological shortcomings of this study, along with the suggestions to strengthen the manuscript.

Reviewers' comments:

Reviewer's Responses to Questions

**Comments to the Author**

1. Is the manuscript technically sound, and do the data support the conclusions?

Reviewer #1: Yes

Reviewer #2: Partly

2. Has the statistical analysis been performed appropriately and rigorously? 

Reviewer #1: Yes

Reviewer #2: No

3. Have the authors made all data underlying the findings in their manuscript fully available?

Reviewer #1: Yes

Reviewer #2: Yes

4. Is the manuscript presented in an intelligible fashion and written in standard English?

Reviewer #1: Yes

Reviewer #2: Yes

5. Review Comments to the Author

Reviewer #1: This is an important and original piece of research addressing an understudied area, namely how the type of wrapper affects KMC delivery and assessment of skin-to-skin practices in the community, post facility discharge.

It is generally a very well written manuscript with a logical and easy to follow structure. The methodology is clear and concise although it would benefit from more detail about randomisation and allocation processes to ensure methods were robust and selection / allocation bias was minimized. There is clear description of the eligibility criteria and well described setting, enabling generalisability of the results. Discussion was relevant and gave a detailed exploration of current evidence base for use of different wrappers and KMC duration data. The article did an excellent job of putting KMC in Malawi into context, with a description of implementation history, showcasing the context of advanced KMC implementation in Malawi, which is not typical of many countries in Sub-Saharan Africa. There was also good discussion of the limitations of self-reporting for KMC duration data although from the supplemental maternal it is clear that direct observation of KMC duration was conducted, yet this is not adequately reported. It would be highly beneficial to expand this section with direct observation data compared with maternal reported data, if possible. Otherwise, issues around missing data and potential ascertainment bias are well described.

The text box on ensuring sustainability and local production is particularly helpful and a novel addition to the study, but would benefit from more detailed description of how the cost per wrapper was derived.

Specific feedback / suggestions for improvement:

Suggest to include study design in the title to make it easier for readers to identify this was a RCT

1. Check the statistics quoted for KMC in the background section. According to Cochrane 2016, mortality risk at 40 – 41 weeks post menstrual age is reduced by 40% (RR0.6) and mortality at latest follow up is reduced by 33% (RR0.67), which is different to what is stated in this article.

2. The primary and secondary outcomes are not clear. It is stated at one stage that the primary outcome is rate of weight gain and then later mentioned that wrap acceptability, duration of STS contact, feeding support and other support were outcomes, with no mention of weight gain. This becomes clearer in the results and discussion when the issue of missing weight outcome data is explained, but for ease of reading it would be beneficial to explain this earlier in the manuscript rather than in the supplemental material.

3. A more detailed description of the statistical analysis would be helpful - What values were considered significant ? Also including 95% confidence intervals is advised.

4. Please give more detail about the randomisation / allocation process – how was the randomisation sequence generated and was this blinded and kept secure? Was allocation concealed and if so, how was it concealed ? What were the timelines involved in screening, allocation and baseline data collection and were they consistent with avoiding selection bias ?

5. It is not clear if the outcome assessors were blinded to the allocation arm, especially for assessing the acceptability outcome. It is possible that unconscious bias may influence how the questions were asked.

6. There are a large (n=75) number of ineligible babies in the “other”: category – in the interests of transparency please provide details of why these babies were excluded.

7. The discussion mentions that data quality issues also affected the results, but it is not clear what this is referring to (? Weight outcome). Please clarify

8. It would be beneficial to include more detail about the pre-enrolment care these babies received, as average age at admission to KMC ward was 1.5 days this suggests they had a short period of stabilization on NICU and then were transferred to KMC. Also, 20% of babies become unwell whilst on KMC ward and it would be informative to know if there is any statistically significant difference between the 2 types of wrappers used, as a marker of safety. Although this information is available in the supplemental material (Table 3a) it is not mentioned in the main manuscript and should be highlighted to the reader.

Reviewer #2: Title: Improving Skin-to-Skin Practice for Babies in Kangaroo Mother Care in Malawi Through the Use of a Customized Baby Wrap

Overall note: Thank you for the opportunity to review this manuscript. It was very well-written and I enjoyed reading the paper as well as learning about this innovative intervention. I believe it is an important contribution to the literature. However, I have several concerns. First, I do not believe the statistical analysis is entirely appropriate; further, critical details are omitted from the methodology section. Some of the text could be organized a little differently to help justify the paper as well. Please see detailed comments below.

Major comments

Abstract

1) Consider adding in that the study was among low birthweight dyads

Introduction

2) To motivate the need for the customized wrap, swap paragraph that starts on line 73 with the one that starts on 64.

3) Ensure that objectives align with outcomes (include breastfeeding, feeding support, family support) and that these outcomes are justified. Why these objectives/outcomes specifically? Acceptability, duration of skin to skin and breastfeeding outcomes make sense (perhaps note that more comfort encourages more use which could mean better outcomes?), but what is the justification for family and social support for KMC and why would that differ by type of wrap (especially food, chores, child-care, etc)?

a. After reading the discussion, there is a need for family support – suggest moving lines 300-304 to intro

Methods

4) Consider noting in study population section that study included multiple births/twins

5) Mention that analyses conducted at two time points- pre and post discharge

6) Please add more detail about how each outcome was assessed. Even if the info is in supplemental file, the outcome variables should be defined up front here, including listing response options.

7) After reviewing supplementary file, it appears the “acceptability” outcomes are binary. The analysis would then be testing the difference in proportion satisfied between the two study groups whereas a t-test is meant to compare the difference in means between groups. Why not use chi-square tests? Similarly, duration of skin-to-skin is reported in categories and the use of a T test is also questionable.

Results

8) Lines 143-146 (first few sentences of the results section) seem to belong in the methods as it is the final sample size.

9) Line 213- were family members’ training part of the study? If so, this should be included in the methods

10) Throughout the text, the ergonomic carrier is called “CarePlus” but in the tables it is “Customized”- consider using “CarePlus” in tables, too.

11) Weight gain is mentioned as the primary outcome of interests, but there is no mention of it until the discussion /limitations. Consider including a line similar to 338 (e.g., “Weight gain results are not included in this study”) then adding some from the supplemental text (lines 27-31) in the results section.

Discussion

12) Line 277-78: there aren’t results related to male involvement in the results section so this statement isn’t supported by the analysis presented. Suggest either adding more to analysis/results, or remove from discussion.

13) Lines 305-311: Suggest moving to results section

a. Also, this is interesting. I suggest finding different headings- it looks too close to an abstract of something that should be an independent paper. Suggest head each paragraph with more simple/plain text-style headings or even questions, e.g. “What was the sustainability assessment?” “What did it show”

14) Lines 321-22: What is the reason to mention that the sample size was based on weight gain in the limitations—this isn’t clear how it could be a limitation. Also, what indications do you have about data quality? This needs more explanation as well.

15) Mention the limitation of generalizability—within Malawi, regionally, etc..

Minor comments

1. (Abstract) Line 31-32: I think the ‘more’ is misplaced. I would suggest saying, “Women using the customized wrap reported being comfortable […] more often than women using the chitenje”

2. Line 50: define/explain “early” discharge

3. Line 53: no apostrophe needed after weeks’; consider adding parentheses around (corrected gestational age)

4. Is there a reference for sentence on line 74-75

5. Figure 3 title add in time point (Day 3)

6. Line 166: delete space between Care and Plus

7. Line 172: restate post-discharge time period/days.

8. Line 188-I don’t see these results specified anywhere- if not, please add “results not shown”

9. Line 189: Reported daily duration – should duration be practice?

10. Supplemental table 3a, please add % to item/variable labels to differentiate from mean (e.g., Without a label, I wasn’t sure at first if died meant number of deaths or %)

11. Table 3: Please add a title for facility based KMC for first few lines. Please add results of provider support to table 3 and also briefly summarize the results that are there.

12. Line 206-07: “frequency of duration of breastfeeding" – I don’t know what this refers to? Are there results to include in the table?

13. Lines 242-249: How similar are other wraps to CarePlus? How similar are traditional wraps in those countries?

14. Line 295: when was the first follow-up visit?

15. Line 298-300: 2013 was several years ago now and much has evolved in newborn/KMC care since then. I would suggest either discussing these findings couched as how things have improved over time or not mentioning it.

16. Line 316-17: I don’t think that is necessary to include (“We had planned…”)

6. PLOS authors have the option to publish the peer review history of their article (what does this mean?). If published, this will include your full peer review and any attached files.

Reviewer #1: No

Reviewer #2: Yes: Lindsay Mallick

---

## [Author Response · Author response to Decision Letter 0]

2 Feb 2020

RESPONSES TO JOURNAL EDITORS

Journal Requirements:

THANK YOU FOR FLAGGING ISSUES OF THE STYLE REQUIREMENTS. WE HAVE MADE MODIFICATIONS ACCORDINGLY.

2.

We note that you have indicated that data from this study are available upon request. PLOS only allows data to be available upon request if there are legal or ethical restrictions on sharing data publicly. For information on unacceptable data access restrictions, please see http://journals.plos.org/plosone/s/data-availability#loc-unacceptable-data-access-restrictions.

WE HAVE ADDED THE ANONYMIZED DATA SET AS A SUPPLEMENTARY FILE. 

3. Please provide additional details regarding participant consent. In the ethics statement in the Methods and online submission information, please ensure that you have specified what type of informed consent you obtained (for instance, written or verbal, and if verbal, how it was documented and witnessed). If your study included minors, state whether you obtained consent from parents or guardians. If the need for consent was waived by the ethics committee, please include this information.

WE HAVE ADDED INFORMATION ON THE TYPE OF CONSENT OBTAINED AND THAT THE STUDY DID NOT INCLUDE MINORS (E.G. ADOLESCENTS WHO WERE MOTHERS). 

Additional Editor Comments (if provided):

Please address the reviewers' comments regarding the methodological shortcomings of this study, along with the suggestions to strengthen the manuscript.

THANK YOU FOR THE OPPORTUNITY TO REVISE AND ADDRESS THE REVIEWER COMMENTS. WE HAVE RESPONDED TO EACH COMMENT AND MADE REVISIONS ACCORDINGLY. 

RESPONSES TO REVIEWER COMMENTS

Reviewer comment AUTHOR RESPONSE

This is an important and original piece of research addressing an understudied area, namely how the type of wrapper affects KMC delivery and assessment of skin-to-skin practices in the community, post facility discharge.

It is generally a very well written manuscript with a logical and easy to follow structure. The methodology is clear and concise although it would benefit from more detail about randomisation and allocation processes to ensure methods were robust and selection / allocation bias was minimized. There is clear description of the eligibility criteria and well described setting, enabling generalisability of the results. Discussion was relevant and gave a detailed exploration of current evidence base for use of different wrappers and KMC duration data. The article did an excellent job of putting KMC in Malawi into context, with a description of implementation history, showcasing the context of advanced KMC implementation in Malawi, which is not typical of many countries in Sub-Saharan Africa. THANK YOU FOR THE ENCOURAGING RESPONSE TO THIS ARTICLE. 

There was also good discussion of the limitations of self-reporting for KMC duration data although from the supplemental maternal it is clear that direct observation of KMC duration was conducted, yet this is not adequately reported. It would be highly beneficial to expand this section with direct observation data compared with maternal reported data, if possible. Otherwise, issues around missing data and potential ascertainment bias are well described.

 THANK YOU FOR YOUR COMMENT. WHILE THE STUDY HAD INTENDED TO CONDUCT DIRECT OBSERVATION OF KMC PRACTICES TO ALLOW FOR THE COMPARISON WITH KMC PRACTICES REPORTED BY MOTHERS, THIS COMPONENT OF THE STUDY WAS NOT ABLE TO BE ADEQUATELY CARRIED OUT DUE TO RESOURCE LIMITATIONS. WE HAVE AMENDED THE SUPPLEMENTAL MATERIAL TO NOTE THIS AND ADDED A POINT TO THE LIMITATIONS 

The text box on ensuring sustainability and local production is particularly helpful and a novel addition to the study, but would benefit from more detailed description of how the cost per wrapper was derived.

 THANK YOU FOR THIS SUPPORT. WE HAVE MADE EDITS TO EXPLAIN HOW WE CALCULATED THE COST.

Suggest to include study design in the title to make it easier for readers to identify this was a RCT

 THANK YOU. WE HAVE MADE THIS CHANGE

1. Check the statistics quoted for KMC in the background section. According to Cochrane 2016, mortality risk at 40 – 41 weeks post menstrual age is reduced by 40% (RR0.6) and mortality at latest follow up is reduced by 33% (RR0.67), which is different to what is stated in this article.

 THANK YOU. WE HAVE MADE THIS CORRECTION.

2. The primary and secondary outcomes are not clear. It is stated at one stage that the primary outcome is rate of weight gain and then later mentioned that wrap acceptability, duration of STS contact, feeding support and other support were outcomes, with no mention of weight gain. This becomes clearer in the results and discussion when the issue of missing weight outcome data is explained, but for ease of reading it would be beneficial to explain this earlier in the manuscript rather than in the supplemental material.

 THANK YOU. WE HAVE ADDED A TABLE DESCRIBING THE OUTCOME INDICATORS. 

3. A more detailed description of the statistical analysis would be helpful - What values were considered significant ? Also including 95% confidence intervals is advised.

 WE HAVE MODIFIED THE ANALYSIS SECTION TO CLARIFY THE P-VALUE USED TO DETERMINE SIGNIFICANCE (P<0.05). 

4. Please give more detail about the randomisation / allocation process – how was the randomisation sequence generated and was this blinded and kept secure? Was allocation concealed and if so, how was it concealed ? What were the timelines involved in screening, allocation and baseline data collection and were they consistent with avoiding selection bias ?

 THANK YOU. WE HAVE REVISED THE TEXT TO PROVIDE MORE DETAILS ON THE RANDOMIZATION / ALLOCATION PROCESS. THE PERSON DOING THE ALLOCATION WAS ONLY AWARE OF THE SUBSEQUENT ALLOCATIONS AND NOT THE INITIAL. THEN INITIAL WAS RANDOM A OR B PICK.

5. It is not clear if the outcome assessors were blinded to the allocation arm, especially for assessing the acceptability outcome. It is possible that unconscious bias may influence how the questions were asked.

 THANK YOU FOR FLAGGING THIS CONCERN. WE HAVE REVISED THE DATA COLLECTION SECTION TO CLARIFY RANDOMIZATION / ALLOCATION PROCESS. 

6. There are a large (n=75) number of ineligible babies in the “other”: category – in the interests of transparency please provide details of why these babies were excluded.

 THE “OTHER” CATEGORY WERE BABIES WITH BIRTHWEIGHT >2000G; HENCE, WE DID NOT INCLUDE THESE BABIES. WE HAVE ADDED A SENTENCE IN THE RESULTS SECTION TO EXPLAIN. 

7. The discussion mentions that data quality issues also affected the results, but it is not clear what this is referring to (? Weight outcome). Please clarify

 WE AGREE THIS IS CONFUSION AND NOT NECESSARY. THUS, WE HAVE REMOVED SENTENCE. 

8. It would be beneficial to include more detail about the pre-enrolment care these babies received, as average age at admission to KMC ward was 1.5 days this suggests they had a short period of stabilization on NICU and then were transferred to KMC. Also, 20% of babies become unwell whilst on KMC ward and it would be informative to know if there is any statistically significant difference between the 2 types of wrappers used, as a marker of safety. Although this information is available in the supplemental material (Table 3a) it is not mentioned in the main manuscript and should be highlighted to the reader.

 THANK YOU FOR THIS COMMENT. OUR STARTING POINT WAS ADMISSION TO KMC. THUS, IT WOULD BE DIFFICULT AT THIS POINT TO INCLUDE PRE-ENROLLMENT CARE AS WE DID NOT DOCUMENT THIS INFORMATION (AVAILABLE IN CASE FILES). AS FOR BABIES WHO BECAME UNWELL IN THE COURSE OF THE STUDY WHILE IN KMC, THERE IS NO STATISTICALLY SIGNIFICANT DIFFERENCE BETWEEN THE 2 TYPES OF WRAPPERS USED. 

REVIEWER #2 

Overall note: Thank you for the opportunity to review this manuscript. It was very well-written and I enjoyed reading the paper as well as learning about this innovative intervention. I believe it is an important contribution to the literature. However, I have several concerns. First, I do not believe the statistical analysis is entirely appropriate; further, critical details are omitted from the methodology section. Some of the text could be organized a little differently to help justify the paper as well. THANK YOU FOR YOUR HELPFUL AND THOROUGH REVIEW. WE HOPE THAT WE HAVE ADEQUATELY ADDRESSED YOUR COMMENTS AND THAT THIS PAPER CAN BE PUBLISHED TO CONTRIBUTE TO THE LITERATURE. 

Abstract

1) Consider adding in that the study was among low birthweight dyads 

Introduction

2) To motivate the need for the customized wrap, swap paragraph that starts on line 73 with the one that starts on 64. THANK YOU FOR THIS SUGGESTION, WE HAVE TAKEN FORWARD THIS RECOMMENDATION. 

3) Ensure that objectives align with outcomes (include breastfeeding, feeding support, family support) and that these outcomes are justified. Why these objectives/outcomes specifically? Acceptability, duration of skin to skin and breastfeeding outcomes make sense (perhaps note that more comfort encourages more use which could mean better outcomes?), but what is the justification for family and social support for KMC and why would that differ by type of wrap (especially food, chores, child-care, etc)?

a. After reading the discussion, there is a need for family support – suggest moving lines 300-304 to intro

 THANK YOU FOR THIS RECOMMENDATION. WE HAVE REVISED THE INTRODUCTION SECTION. 

Methods

4) Consider noting in study population section that study included multiple births/twins

 WE HAVE ADDED THIS. 

5) Mention that analyses conducted at two time points- pre and post discharge

 WE HAVE ADDED THIS. 

6) Please add more detail about how each outcome was assessed. Even if the info is in supplemental file, the outcome variables should be defined up front here, including listing response options.

 THANKS FOR YOUR SUGGESTION. WE HAVE INCLUDED A TABLE WITH THE DEFINITIONS OF THE SECONDARY OUTCOMES (THE MAIN FOCUS OF THIS PAPER) AS TABLE 2 IN THE METHODS SECTION.

7) After reviewing supplementary file, it appears the “acceptability” outcomes are binary. The analysis would then be testing the difference in proportion satisfied between the two study groups whereas a t-test is meant to compare the difference in means between groups. Why not use chi-square tests? Similarly, duration of skin-to-skin is reported in categories and the use of a T test is also questionable.

 THANK YOU FOR THIS COMMENT. WE DID INDEED USE THE CHI SQUARE TEST FOR CATEGORICAL OUTCOMES (THE T-TEST WAS USED FOR COMPARING THE WEIGHT GAIN, A CONTINUOUS VARIABLE AND THE PRIMARY OUTCOME WHICH WAS NOT THE FOCUS OF THIS PAPER) AND HAD MADE AN ERROR IN THE METHODS DESCRIPTION. WE HAVE MODIFIED THE METHODS ACCORDINGLY.

Results

8) Lines 143-146 (first few sentences of the results section) seem to belong in the methods as it is the final sample size THANK YOU FOR THIS SUGGESTION. THE METHODS SECTION EXPLAINS HOW THE SAMPLE SIZE WOULD BE COLLECTED; AND THUS THE RESULTS SECTION SHOULD INCLUDE AN EXPLANATION OF THE PROCESS AND RESULTS OF THE SAMPLE SIZE. 

9) Line 213- were family members’ training part of the study? If so, this should be included in the methods

 TRAINING FAMILY MEMBERS WAS NOT PART OF OUR STUDY BUT RATHER STANDARD PRACTICE AT THE FACILITIES. WE ONLY ASKED MOTHERS WHETHER A FAMILY MEMBER WAS PART OF THE TRAINING. 

10) Throughout the text, the ergonomic carrier is called “CarePlus” but in the tables it is “Customized”- consider using “CarePlus” in tables, too.

 THANK YOU FOR FLAGGING THIS INCONSISTENCY. WE HAVE CHANGED TO “CAREPLUS” THROUGHOUT THE PAPER. 

11) Weight gain is mentioned as the primary outcome of interests, but there is no mention of it until the discussion /limitations. Consider including a line similar to 338 (e.g., “Weight gain results are not included in this study”) then adding some from the supplemental text (lines 27-31) in the results section.

 THANK YOU FOR THIS SUGGESTION. WE HAVE MADE EDITS ACCORDINGLY. 

Discussion

12) Line 277-78: there aren’t results related to male involvement in the results section so this statement isn’t supported by the analysis presented. Suggest either adding more to analysis/results, or remove from discussion.

 FIGURE 3 PRESENTS RESULTS ON MALE ACCEPTABILITY. WE HAVE ADDED A SENTENCE IN THE SECTION ON FAMILY SUPPORT TO HIGHLIGHT THAT HUSBANDS WERE COMMONLY MENTIONED AS PROVIDING SUPPORT AS WELL AS MOTHERS AND SISTERS.

13) Lines 305-311: Suggest moving to results section

a. Also, this is interesting. I suggest finding different headings- it looks too close to an abstract of something that should be an independent paper. Suggest head each paragraph with more simple/plain text-style headings or even questions, e.g. “What was the sustainability assessment?” “What did it show”

 THANK YOU FOR THESE SUGGESTIONS. WE HAVE IMPLEMENTED THESE CHANGES. 

14) Lines 321-22: What is the reason to mention that the sample size was based on weight gain in the limitations—this isn’t clear how it could be a limitation. Also, what indications do you have about data quality? This needs more explanation as well.

 WE AGREE THIS IS CONFUSION AND NOT NECESSARY. THUS, WE HAVE REMOVED SENTENCE. 

15) Mention the limitation of generalizability—within Malawi, regionally, etc..

 WE HAVE INCLUDED THIS IN THE LIMITATIONS SECTION OF THE DISCUSSION.

Minor comments

 THANK YOU FOR THESE COMMENTS. ALL HAVE BEEN ADDRESSED.

1. (Abstract) Line 31-32: I think the ‘more’ is misplaced. I would suggest saying, “Women using the customized wrap reported being comfortable […] more often than women using the chitenje” EDITED

2. Line 50: define/explain “early” discharge REVISED SENTENCE. 

3. Line 53: no apostrophe needed after weeks’; consider adding parentheses around (corrected gestational age) EDITED. 

4. Is there a reference for sentence on line 74-75 ADDED REFERENCE. 

5. Figure 3 title add in time point (Day 3) 

6. Line 166: delete space between Care and Plus EDITED.

7. Line 172: restate post-discharge time period/days. ADDED POST DISCHARGE TIME

8. Line 188-I don’t see these results specified anywhere- if not, please add “results not shown” WE HAVE REMOVED THIS SENTENCE. 

9. Line 189: Reported daily duration – should duration be practice? WE MADE MINOR EDIT TO CLARIFY SENTENCE STRUCTURE. 

10. Supplemental table 3a, please add % to item/variable labels to differentiate from mean (e.g., Without a label, I wasn’t sure at first if died meant number of deaths or %) WE HAVE ADDED THE STATUS AT DISCHARGE FROM KMC (%) TO THE SUPPLEMENTARY FILE

11. Table 3: Please add a title for facility based KMC for first few lines. Please add results of provider support to table 3 and also briefly summarize the results that are there. WE HAVE CORRECTED THE TABLE TO CLEARLY SHOW DATA FROM FACILITY-KMC AND POST-DISCHARGE.

12. Line 206-07: “frequency of duration of breastfeeding" – I don’t know what this refers to? Are there results to include in the table? WE REMOVED THIS SENTENCE. 

13. Lines 242-249: How similar are other wraps to CarePlus? How similar are traditional wraps in those countries? WE DID NOT CONDUCT A COMPARATIVE ANALYSIS IN TERMS OF TEXTURE OR TYPE OF WRAPS WITH THESE STUDIES AS THIS INFORMATION WAS NOT ALWAYS PROVIDED AND OUT OF THE SCOPE OF THIS RESEARCH. WE, THUS, RECOMMEND FUTURE RESEARCH SHOULD CONSIDER INCLUDING DETAILS ON THE TYPE OF WRAP AS WELL AS TEST DIFFERENT TYPES OF WRAPS.

14. Line 295: when was the first follow-up visit? THIS HAS BEEN ADDED

15. Line 298-300: 2013 was several years ago now and much has evolved in newborn/KMC care since then. I would suggest either discussing these findings couched as how things have improved over time or not mentioning it. THANK YOU. WE HAVE MOVED TO INTRODUCTION AND REVISED THE FRAMING.

16. Line 316-17: I don’t think that is necessary to include (“We had planned…”) WE HAVE REVISED THIS.

---

## [Editor Report · Decision Letter 1]

13 Feb 2020

Improving Skin-to-Skin Practice for babies in Kangaroo Mother Care in Malawi through the use of a customized baby wrap: a randomized control trial

PONE-D-19-24170R1

Dear Dr. Chavula,

We are pleased to inform you that your manuscript has been judged scientifically suitable for publication and will be formally accepted for publication once it complies with all outstanding technical requirements.

With kind regards,

Jennifer Yourkavitch

Academic Editor

PLOS ONE
---

## [Editor Report · Acceptance letter]

3 Mar 2020

PONE-D-19-24170R1 

Improving Skin-to-Skin Practice for babies in Kangaroo Mother Care in Malawi through the use of a customized baby wrap: a randomized control trial 

Dear Dr. Chavula:

I am pleased to inform you that your manuscript has been deemed suitable for publication in PLOS ONE. Congratulations! Your manuscript is now with our production department. 

With kind regards,

on behalf of

Dr. Jennifer Yourkavitch 

Academic Editor

PLOS ONE